# Clinical Significance of Preoperative Hematological Parameters in Patients with D2-Resected, Node-Positive Stomach Cancer

**DOI:** 10.3390/biomedicines10071565

**Published:** 2022-06-30

**Authors:** Jun Su Park, Jeong Il Yu, Do Hoon Lim, Heerim Nam, Young Il Kim, Jeeyun Lee, Won Ki Kang, Se Hoon Park, Seung Tae Kim, Jung Yong Hong, Tae Sung Sohn, Jun Ho Lee, Ji Yeong An, Min Gew Choi, Jae Moon Bae

**Affiliations:** 1Department of Radiation Oncology, Chungnam National University Sejong Hospital, Chungnam National University School of Medicine, Sejong 30099, Korea; jsrtpark@cnuh.co.kr (J.S.P.); minesota@cnuh.co.kr (Y.I.K.); 2Department of Radiation Oncology, Samsung Medical Center, Sungkyunkwan University School of Medicine, Seoul 06351, Korea; dh8.lim@samsung.com; 3Department of Radiation Oncology, Kangbuk Samsung Hospital, Sungkyunkwan University School of Medicine, Seoul 03181, Korea; heerim.nam@samsung.com; 4Department of Medicine, Division of Hematology-Oncology, Samsung Medical Center, Sungkyunkwan University School of Medicine, Seoul 06351, Korea; jyun.lee@samsung.com (J.L.); wonki.kang@samsung.com (W.K.K.); sh1767.park@samsung.com (S.H.P.); seungtae1.kim@samsung.com (S.T.K.); jungyong.hong@samsung.com (J.Y.H.); 5Department of Surgery, Samsung Medical Center, Sungkyunkwan University School of Medicine, Seoul 06351, Korea; ts.sohn@samsung.com (T.S.S.); junho3371.lee@samsung.com (J.H.L.); jar319.an@samsung.com (J.Y.A.); mingew.choi@samsung.com (M.G.C.); jmoon.bae@samsung.com (J.M.B.)

**Keywords:** stomach neoplasm, prognosis, leukocyte count, platelet count, radiotherapy, chemotherapy

## Abstract

The purpose of the present study was to investigate the clinical significance of preoperative hematological parameters in patients with advanced stomach cancer, and to explore who might benefit from adjuvant concurrent chemoradiotherapy (CCRT) compared to chemotherapy alone. Among 1032 patients with node-positive stomach cancer who had a confirmed diagnosis after complete D2 resection, and who received adjuvant chemotherapy alone or CCRT, a total of 692 patients was selected using propensity score matching. Among absolute neutrophil count, absolute lymphocyte count (ALC), absolute monocyte count (AMC), platelet count, neutrophil-to-lymphocyte ratio, lymphocyte-to-monocyte ratio, and platelet-to-lymphocyte ratio, AMC was the most relevant prognostic factor for overall survival and recurrence-free survival (hazard ratio (HR) 1.674, 95% confidence interval (CI) 1.180–2.376; HR 1.908, 95% CI 1.650–2.695, respectively). In a subgroup with a high ALC, patients treated with adjuvant CCRT had a favorable recurrence-free survival (HR 0.620, 95% CI 0.393–0.980) compared to those treated with chemotherapy alone. Further study is needed to confirm our findings and to develop tailored adjuvant treatment.

## 1. Introduction

Stomach cancer is the fifth most common cancer (5.6% of all cases) and the fourth leading cause of death (7.7% of all cancer-related deaths) based on the Global Cancer Statistics 2020 [1]. The incidence of stomach cancer is the highest in Eastern Asia—especially in Japan and Mongolia—and Eastern Europe.

Based on the results of two large phase III randomized controlled trials (RCTs) conducted on Asian patients, the Korean Gastric Cancer Association recommends adjuvant chemotherapy with S-1 alone or capecitabine plus oxaliplatin for patients with pathological stage II or III stomach cancer [2,3]. The survival benefit from both chemotherapy regimens was confirmed by assessing the 5-year follow-up data [4,5]. On the other hand, the role and necessity of adjuvant concurrent chemoradiotherapy (CCRT) in completely D2-resected gastric cancer remains controversial. The Intergroup Study 0116 (INT-0116), which showed that adjuvant CCRT improved overall survival (OS) and relapse-free survival when compared with surgery alone, was criticized because only 10% of patients had undergone a D2 resection, which is considered a standard surgical procedure for advanced stomach cancer in Asian countries [6]. Additionally, a phase III RCT conducted in Korea compared the treatment outcomes between adjuvant chemotherapy alone and CCRT after D2 resection. CCRT reduced locoregional recurrence, but there was no benefit of OS in the CCRT group [7,8,9].

Since blood testing is an essential examination before gastric surgery, preoperative hematological parameters are easily available, and might be useful for predicting prognosis along with various tumor characteristics. There is a growing body of evidence demonstrating the prognostic value of hematological parameters—including neutrophils, lymphocytes, and monocytes—as well as the ratios of these cells—including neutrophil-to-lymphocyte ratio (NLR), lymphocyte-to-monocyte ratio (LMR), and platelet-to-lymphocyte ratio (PLR)—in various cancers [10,11,12,13]. These hematological parameters are related to the innate immune system or adaptive immune response against pathogens, and are involved in the prevention, carcinogenesis, exacerbation, and metastasis of cancer. The association between these hematological parameters and the extent of cancer and its biology suggests the possibility of an association with the prognosis as well as the risk and pattern of recurrence in several cancers [14,15,16]. Therefore, it is possible to provide important information for implementing customized adjuvant treatment in patients with completely resected gastric cancer.

In this study, we aimed to investigate the prognostic significance of preoperative hematological parameters in patients with D2-resected, node-positive stomach cancer. We also explored who benefited more from adjuvant CCRT than from chemotherapy alone.

## 2. Materials and Methods

### 2.1. Patients

Between April 2008 and February 2013, a total of 1032 patients with D2-resected, node-positive stomach cancer received either adjuvant chemotherapy or adjuvant chemoradiotherapy (CCRT) at Samsung Medical Center. Most of these patients (*n* = 1005) received adjuvant treatment according to the protocol of the Adjuvant Chemotherapy Trial of TS-1 for Gastric Cancer (ACTS-GC) and the Intergroup Study 0116 (INT-0116) [2,6]. The other 21 patients received adjuvant chemotherapy according to the protocol of the capecitabine and oxaliplatin adjuvant study in stomach cancer (CLASSIC), and 6 patients received off-protocol chemotherapy. This study included patients who received adjuvant treatment according to the ACTS-GC or INT-0116 protocols.

### 2.2. Treatment

Adjuvant treatment, which can be CCRT for 5 months or chemotherapy alone for 1 year, was determined based on patient preference, with informed consent. Patients who chose chemotherapy alone followed the ACTS-GC protocol, and received two oral doses of 40 mg/m^2^ S^−1^ per day for 4 weeks, followed by 2 weeks of rest [2]. This 6-week cycle of chemotherapy was repeated eight times during the first year after surgery. CCRT was performed according to the INT-0116 study protocol [6]. Six cycles of chemotherapy consisted of 425 mg/m^2^ fluorouracil per day and 20 mg/m^2^ leucovorin per day, taken for 5 days. The second and third cycles were administered on the first four days and the last three days of radiotherapy, respectively, and the fluorouracil dose was reduced to 400 mg/m^2^ per day. A total radiation dose of 45 Gy was delivered in daily fractions of 1.8 Gy over a duration of 5 weeks. The radiation field encompassed the tumor bed, anastomosis site, duodenal stump, regional lymph nodes, and 2 cm beyond the proximal and distal resection margins. The remnant stomach was not routinely encompassed in the radiation field.

### 2.3. Preoperative Blood Cell Counts and Ratios

Complete blood counts with a differential count within 60 days before surgery were used to calculate blood cell ratios. Seven patients who had unavailability of data were excluded from this study. NLR was defined as the neutrophil count divided by the lymphocyte count. LMR was defined as the lymphocyte count divided by the monocyte count. PLR was defined as the platelet count divided by the lymphocyte count. Continuous variables were converted into categorical variables based on the median values.

### 2.4. Evaluation and Statistical Analysis

The chi-squared test, independent *t*-test, and one-way analysis of variance were used to compare patient characteristics between the adjuvant treatment groups. OS was defined as the time from the date of surgery to the date of death from any cause. Recurrence-free survival (RFS) was defined as the time from the date of surgery to the date of recurrence or death—whichever occurred first. Kaplan–Meier survival curves were constructed and compared using the log-rank test to investigate the association between patient characteristics and OS and RFS. We also conducted a Cox regression analysis using forward stepwise regression to evaluate the effects of patient characteristics on OS and RFS. Statistically significant covariates in the univariate analyses and the type of adjuvant treatment, which was our interest, were included in the multivariate analysis. Statistical significance was set at *p* < 0.05. Statistical analyses were performed using SPSS, version 26 (IBM, Armonk, NY, USA). Propensity score matching (PSM) was performed between the CCRT group and the chemotherapy-alone group using R software, version 4.1.0 (Foundation for Statistical computing, Vienna, Austria). Age, sex, T stage, N stage, stage group, Lauren classification, and surgical extent were included in the calculation of propensity scores.

### 2.5. Ethical Statement

This study was approved by the Institutional Review Board of Samsung Medical Center (SMC 2022-03-104-001).

## 3. Results

### 3.1. Patient Characteristics

After PSM matching, a total of 692 patients was included in the study, and their characteristics are shown in Table 1. The mean patient age was 55.0 years (95% confidence interval (CI) 54.2–55.9 years old). All patients underwent D2 resection with negative margins, and had lymph node metastases. The mean number of harvested lymph nodes was 47 (range, 16–119). No patient received neoadjuvant treatment prior to surgery. The proportions of patients with stages I, II, and III were 10.1%, 42.2%, and 47.7%, respectively. Stage was classified according to the seventh edition of the American Joint Committee on Cancer’s staging system. The mean interval between surgery and initiation of adjuvant treatment was 5 weeks (range: 2–15 weeks). Patients’ characteristics before matching are presented in Appendix A.

### 3.2. Preoperative Blood Cell Counts and Ratios

The mean values of absolute neutrophil count (ANC), absolute lymphocyte count (ALC), absolute monocyte count (AMC), platelet count (PC), NLR, LMR, and PLR were 3731.8/µL (95% CI 3620.9–3842.6/µL), 2130.7/µL (95% CI 2083.2–2178.2/µL), 454.0/µL (95% CI 440.9–467.1/µL), 248.8 × 10^3^/µL (95% CI 244.0–253.5 × 10^3^/µL), 1.920 (95% CI 1.824–2.015), 5.184 (95% CI 5.034–5.334), and 128.0 (95% CI 123.7–132.3), respectively. The CCRT group had a higher mean ALC (2188.7/µL versus 2072.7/µL, *p* = 0.016), a lower NLR (1.796 versus 2.044, *p* = 0.004), and a lower PLR (122.6 versus 133.4, *p* = 0.002) than the chemotherapy-alone group. The differences in ANC, AMC, PC, and LMR between the two groups were not statistically significant. The median values of ANC, ALC, AMC, PC, NLR, LMR, and PLR were 3448.5/µL, 2074.4/µL, 422.5/µL, 240.5 × 10^3^/µL, 1.657, 4.956, and 117.6, respectively. Based on the median values, the CCRT group had more patients with low NLR and high LMR compared to the chemotherapy-alone group (Table 1).

Figure 1 shows the differences in the mean values of hematological parameters according to the disease stage. The mean values of ANC and NLR tended to be higher and the mean ALC tended to be lower in patients with an advanced disease stage, but the differences were not statistically significant. The mean values of AMC, PC, and PLR were higher in patients with stage III disease than in those with stage IB–II disease (469.4/µL versus 440.1/µL, *p* = 0.028; 261.0 × 10^3^/µL versus 237.6 × 10^3^/µL, *p* < 0.001; 137.5 versus 119.3, *p* < 0.001, respectively). On the other hand, the mean LMR was lower in patients with stage III disease than in those with stage IB–II disease (4.944 versus 5.403, *p* = 0.003).

### 3.3. Survival Results

The median follow-up time was 79 months (IQR 60–95 months). The mean OS and RFS were 103.3 months (95% CI 100.6–106.1 months) and 98.7 months (95% CI 95.4–102.0 months), respectively. The 5-year OS and RFS were 83.4% and 80.6%, respectively. According to the results of the log-rank test, age > 65 years, advanced T stage, advanced N stage, advanced stage group, TG, high AMC, and low LMR were associated with worse OS (Table 2). In terms of RFS, age > 65 years, advanced T stage, advanced N stage, advanced stage group, TG, high AMC, and low LMR were associated with worse RFS. The type of adjuvant treatment did not affect OS or RFS.

We evaluated the effects of patient characteristics on OS and RFS using univariate Cox regression analysis, and the results are demonstrated in Table 3. Multivariate Cox regression analysis showed that age >65 years (HR 2.190, 95% CI 1.531–3.132), advanced T stage (HR 2.099, 95% CI 1.261–3.492 for T3; HR 4.346, 95% CI 2.587–7.300 for T4), advanced N stage (HR 1.770, 95% CI 1.081–2.896 for N2; HR 2.340, 95% CI 1.436-3.814 for N3), TG (HR 1.548, 95% CI 1.094–2.191), and AMC > 422.5/µL (HR 1.674, 95% CI 1.180–2.376) were significant prognostic factors for worse OS (Table 4). Age >65 years (HR 1.444, 95% CI 1.001–2.084), advanced T stage (HR 2.509, 95% CI 1.521–4.141 for T3; HR 4.717, 95% CI 2.829–7.865 for T4), advanced N stage (HR 2.496, 95% CI 1.500–4.155 for N2; HR 3.591, 95% CI 2.150–5.999 for N3), and AMC > 422.5/µL (HR 1.908, 95% CI 1.650–2.695) were significant prognostic factors for worse RFS. While LMR and AMC were both closely associated with OS and RFS in univariate analysis, only AMC remained significant in multivariate analysis. Figure 2 shows the adjusted OS and RFS curves according to AMC.

### 3.4. Subgroup Analysis

Multivariate analysis was repeated for subgroups categorized according to blood cell counts and ratios. As with the results of the analysis of all patients (*n* = 692), T stage and either N stage or stage group were found to be valid prognostic factors for OS and RFS in all subgroups. The type of adjuvant treatment did not affect the treatment outcomes in any subgroup, except for one: in the subgroup with high ALC (>2074.0/µL), patients who received adjuvant CCRT had a favorable RFS compared to those who received chemotherapy alone (HR 0.620, 95% CI 0.393–0.980, *p* = 0.040) (Table 5 and Figure 3).

## 4. Discussion

Our results indicate that preoperative blood cell counts and their ratios are associated with the stage of stomach cancer. ANC, AMC, PC, NLR, and PLR were higher in patients with advanced disease. By contrast, ALC and LMR were lower in patients with advanced disease. Among these factors, AMC was the only independent prognostic factor for both OS and RFS. Subgroup analyses revealed that patients with a high ALC benefited from adjuvant CCRT rather than from chemotherapy alone.

The negative prognostic impact of preoperative AMC on OS and RFS in the present study was similar to that reported in previous studies. Low ALC and high AMC have been reported as prognostic factors for OS and disease-free survival (DFS) in resectable stomach cancer, with cutoff values of 1734/µL and 1720/µL, and 672.4/µL and 510/µL, respectively [17,18]. The prognostic significance of NLR, LMR, and PLR in stomach cancer has been reported in several meta-analyses (Table 6) [13,19,20,21,22]. Although the target population and cutoff values of each study were widely heterogeneous, an increase in NLR and PLR and a decrease in LMR were associated with a worse prognosis. Contrary to the results of those studies, the prognostic significance of NLR and PLR was not found in the present study. While the LMR was closely associated with OS and RFS in univariate analysis, it lost its significance in multivariate analysis owing to AMC. This finding might have been obtained because the subjects of the present study were relatively healthy and patients with early-stage disease were able to receive curative surgical resection and adjuvant treatment. In fact, the mean values of NLR, LMR, and PLR in the present study, which were 1.920 (95% CI 1.824–2.015), 5.184 (95% CI 5.034–5.334), and 128.0 (95% CI 123.7–132.3), respectively, were lower than the reference values suggested by Schiefer et al., which were 4.5, 5.43, and 152, respectively—especially for NLR [23]. Additionally, our values were not significantly different from those of 12,160 healthy Korean populations obtained from the checkup center of a tertiary hospital: NLR of 1.65 (95% CI 0.107–3.193), LMR of 5.31 (95% CI 2.008–8.612), and PLR of 132.40 (95% CI 46.764–218.006) [24].

The underlying mechanism of the prognostic significance of preoperative blood cell counts and their ratios can be explained by the role of each blood cell. The pro-cancer properties of neutrophils, monocytes, and platelets have been reported in many studies. Neutrophils are innate immune cells that also play a role in cancer’s initiation, progression, and metastasis [25]. Neutrophils recruited to inflammatory sites promote cancer initiation by increasing deoxyribonucleic acid (DNA) damage, angiogenesis, and immunosuppression through reactive oxygen species, reactive nitrogen species, and angiogenic factors. Growth factors released by neutrophils and neutrophil extracellular traps (NETs) promote cancer progression. Cytokines and granular proteins released by neutrophils and NETs are also involved in each step of cancer metastasis, including preparation of a pre-metastatic niche, angiogenesis, cancer cell migration, intravasation, extravasation, and survival of cancer cells in the peripheral blood. Cancer cells and stroma release various cytokines that reprogram neutrophils to function as pro-cancer neutrophils. Monocytes are recruited and differentiated into tumor-associated macrophages (TAMs) in response to the tumor microenvironment [26]. TAMs produce various cytokines that promote cancer progression and inhibit cytotoxic T-lymphocyte responses through the programmed death-1/programmed death-ligand-1 signaling pathway and induction of regulatory T cells. Platelet activation by cancer also exerts pro-cancer effects, including stimulating tumor growth, preparing the metastatic niche, and helping the metastatic cells to survive in circulation [27]. Lymphocytes play a central role in the anticancer immune response, and a recent meta-analysis indicated that a low blood lymphocyte count before treatment among patients with solid tumors was associated with worse OS and progression-free survival [28]. The mechanism underlying lymphocytopenia in solid tumors has not been clearly elucidated; however, it is widely believed that lymphocytopenia may occur as a result of cancer-induced immune suppression and increased lymphocyte apoptosis. Based on these immunological mechanisms, increased activation of neutrophils, monocytes, and platelets by tumors and low lymphocyte counts—in other words, high NLR and PLR, and low LMR—could be used as prognostic factors for worse prognosis.

Our finding that there is a specific subgroup that benefits more from CCRT than from chemotherapy alone, even after D2 resection, could represent a significant concept for radiation oncologists. A well-designed phase III RCT was conducted to investigate the role of adjuvant CCRT in patients who received D2 resection, but there were no differences in DFS and OS between the chemotherapy-alone and the CCRT groups [8]. Subgroup analysis indicated that regional recurrence was lower in the CCRT group than in the chemotherapy-alone group (5/230 patients versus 23/228 patients, *p* < 0.001), and node-positive patients benefited more from CCRT than other subgroups [9]. However, in a subsequent phase III RCT that included only node-positive patients, DFS in CCRT with S-1 and oxaliplatin was not superior compared to that in doublet chemotherapy alone (HR 0.971, *p* = 0.879) [29]. Although there is insufficient evidence to support adjuvant CCRT after D2 resection, there is a clear benefit in decreasing locoregional recurrence from CCRT. Thus, further studies that discuss not only conventional tumor staging, but also molecular aspects of tumors, are necessary to identify candidates for CCRT.

Recently, microsatellite instability (MSI) status—one of the molecular aspects—has become a spotlight as a prognostic factor and a predictive factor for systemic therapy in resectable stomach cancer. A meta-analysis of four RCTs reported that MSI was associated with longer OS and DFS (HR 1.78, 95% CI 1.17–2.73 and HR 1.88, 95% CI 1.28–2.76, respectively) [30]. It also indicated a potential lack of benefit of perioperative or adjuvant chemotherapy for patients with high-MSI disease (HR 1.50, 95% CI 0.55–4.12 for OS; HR 1.27, 95% CI 0.53–3.04 for DFS), while patients with low-MSI or microsatellite-stable disease benefited from chemotherapy in terms of OS and DFS (HR 0.75, 95% CI 0.60–0.94; and HR 0.65, 95% CI 0.53–0.79, respectively). Although it was a study on advanced endometrial cancer, patients with high-MSI disease had an improved 2-year progression-free survival with adjuvant CCRT compared with that of chemotherapy alone (40.0% versus 29.5%, *p* = 0.04) [31]. The authors believed that the deficiency in mismatch repair could enhance the response to radiation-induced DNA damage.

We compared patient characteristics between the subgroups categorized based on their mean ALC values to explain the benefit of CCRT in the high-ALC group (Appendix A). There were no significant differences in the proportions of patients in terms of age > 65 years, T stage, N stage, stage group, Lauren classification, surgical extent, and type of adjuvant treatment. By contrast, the high-ALC group had lower proportions of patients with high NLR (29.6% versus 70.0%, *p* < 0.001) and high PLR (24.1% versus 75.8%, *p* < 0.001), and a higher proportion of patients with high LMR (64.6% versus 35.4%, *p* < 0.001), which represents intact host immunity and low levels of systemic inflammation. Radiation-induced immunogenic cell death can activate adaptive immune response through tumor antigen presentation to cytotoxic T lymphocytes in immunocompetent individuals [32].

This study has several limitations. Our results cannot be applied to node-negative patients, even if they have an advanced T stage, because we included only node-positive patients. Second, there were significant differences in patient characteristics between the different adjuvant treatment groups before PSM, owing to the retrospective study design. We observed tendencies whereby elderly patients preferred chemotherapy alone owing to the convenience of oral chemotherapy, while patients with advanced disease preferred adjuvant CCRT despite the inconvenience of daily visits for radiotherapy. The cutoff values of continuous variables can affect the results—especially those of NLR and LMR, because they are very small compared with those of ALC and PLR.

## 5. Conclusions

AMC, together with T and N stages, was a prognostic factor for OS and RFS in patients with D2-resected, node-positive stomach cancer. Patients with a high AMC had a worse prognosis than those with a low AMC. There were no differences in treatment outcomes according to adjuvant treatment in most patients. However, in a subgroup with high ALC, patients who received adjuvant CCRT had a favorable RFS compared to those who received adjuvant chemotherapy alone. Thus, preoperative hematological parameters might be useful not only as prognostic factors, but also as predictive factors. Further study is needed to confirm our findings and to develop tailored adjuvant treatments.

## Figures and Tables

**Figure 1 biomedicines-10-01565-f001:**
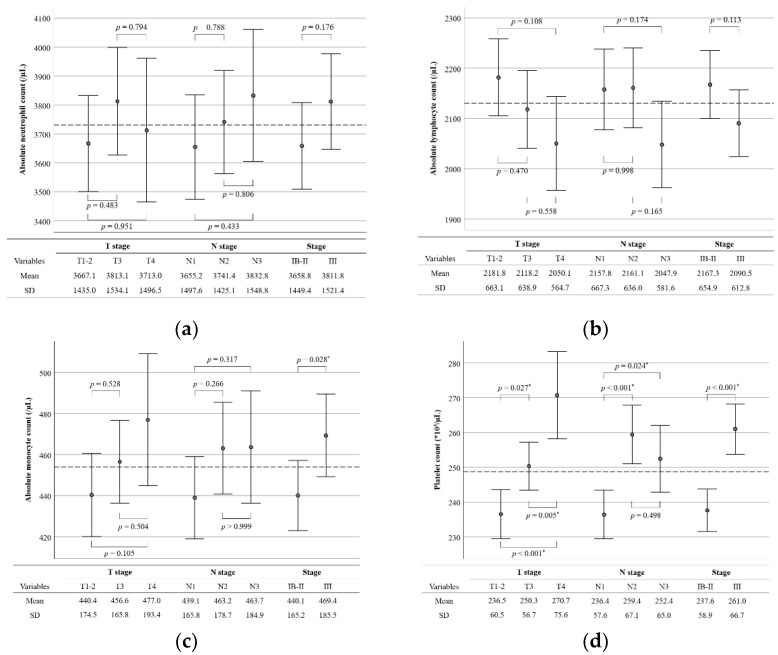
Error bar plots comparing preoperative blood cell counts and their ratios according to tumor stage: (**a**) absolute neutrophil count; (**b**) absolute lymphocyte count; (**c**) absolute monocyte count; (**d**) platelet count; (**e**) neutrophil-to-lymphocyte ratio; (**f**) lymphocyte-to-monocyte ratio; (**g**) platelet-to-lymphocyte ratio. Asterisks (*) indicate *p* < 0.005. The dashed line indicates the mean value from all patients (*n* = 692). Circles and lines indicate the mean values and 95% confidence intervals in each subgroup, respectively. SD, standard deviation.

**Figure 2 biomedicines-10-01565-f002:**
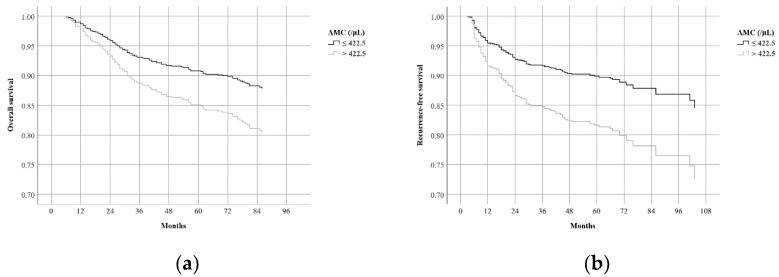
Adjusted survival curves according to AMC in all patients (*n* = 692): (**a**) Overall survival curve; (**b**) recurrence-free survival curve. AMC, absolute monocyte count.

**Figure 3 biomedicines-10-01565-f003:**
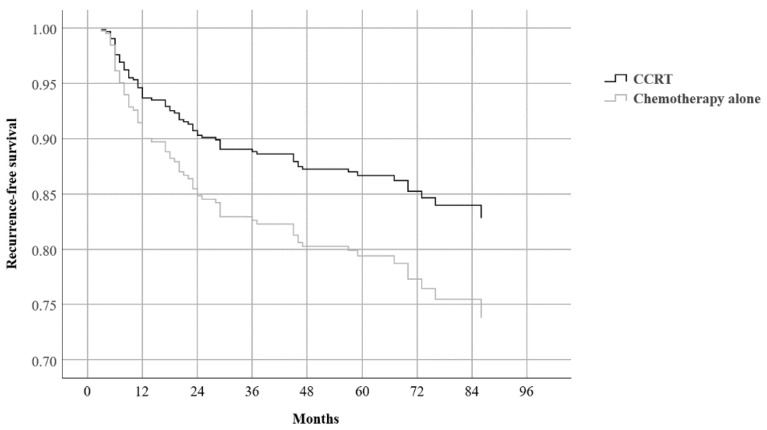
Adjusted recurrence-free survival curve according to the type of adjuvant treatment in patients with an absolute lymphocyte count > 2074.0/µL (*n* = 345). CCRT, concurrent chemoradiotherapy.

**Table 1 biomedicines-10-01565-t001:** Patients’ characteristics.

Variables	All Patients (*n* = 692)	CCRT (*n* = 346)	Chemotherapy Alone (*n* = 346)	*p*-Value
Age				
≤65	550 (79.5%)	282 (81.5%)	268 (77.5%)	0.188
>65	142 (20.5%)	64 (18.5%)	78 (22.5%)	
Sex				
Male	417 (60.3%)	216 (62.4%)	201 (58.1%)	0.244
Female	275 (39.7%)	130 (37.6%)	145 (41.9%)	
T stage				
T1–2	288 (41.6%)	147 (42.5%)	141 (40.8%)	0.820
T3	262 (37.9%)	127 (36.7%)	135 (39.0%)	
T4	142 (20.5%)	72 (20.8%)	70 (20.2%)	
N stage				
N1	266 (38.4%)	132 (38.2%)	134 (38.7%)	0.788
N2	248 (35.8%)	128 (37.0%)	120 (34.7%)	
N3	178 (25.7%)	86 (24.9%)	92 (26.6%)	
Stage				
IB–II	362 (52.3%)	182 (52.6%)	180 (52.0%)	0.879
III	330 (47.7%)	164 (47.4%)	166 (48.0%)	
Lauren classification				
Non-intestinal	458 (66.2%)	228 (65.9%)	230 (66.5%)	0.872
Intestinal	234 (33.8%)	118 (34.1%)	116 (33.5%)	
Surgical extent				
STG	479 (69.2%)	242 (69.96%)	237 (68.5%)	0.681
TG	213 (30.8%)	104 (30.1%)	109 (31.5%)	
ANC (/µL)				
≤3448.5	346 (50.0%)	172 (49.7%)	174 (50.3%)	0.879
>3448.5	346 (50.0%)	174 (50.3%)	172 (49.7%)	
ALC (/µL)				
≤2074.0	347 (50.1%)	163 (47.1%)	184 (53.2%)	0.110
>2074.0	345 (49.9%)	183 (52.9%)	162 (46.8%)	
AMC (/µL)				
≤422.5	346 (50.0%)	174 (50.3%)	172 (49.7%)	0.879
>422.5	346 (50.0%)	172 (49.7%)	174 (50.3%)	
PC (×10^3^/µL)				
≤240.5	346 (50.0%)	170 (49.1%)	176 (50.9%)	0.648
>240.5	346 (50.0%)	176 (50.9%)	170 (49.1%)	
NLR				
≤1.657	347 (50.1%)	188 (54.3%)	159 (46.7%)	0.027 *
>1.657	345 (49.9%)	158 (45.7%)	187 (54.0%)	
LMR				
≤4.956	346 (50.0%)	159 (46.0%)	187 (54.0%)	0.033 *
>4.956	346 (50.0%)	187 (54.0%)	159 (46.0%)	
PLR				
≤117.6	346 (50.0%)	180 (52.0%)	166 (48.0%)	0.287
>117.6	346 (50.0%)	166 (48.0%)	180 (52.0%)	

CCRT, concurrent chemoradiotherapy; STG, subtotal gastrectomy; TG, total gastrectomy; ANC, absolute neutrophil count; ALC, absolute lymphocyte count; AMC, absolute monocyte count; PC, platelet count; NLR, neutrophil-to-lymphocyte ratio; LMR, lymphocyte-to-monocyte ration; PLR, platelet-to-lymphocyte ratio. Asterisks (*) indicate *p* < 0.005.

**Table 2 biomedicines-10-01565-t002:** Comparison of OS and RFS according to patient characteristics.

Variables	5-Year OS	*p*-Value	5-Year RFS	*p*-Value
Age				
≤65	87.5%	<0.001 *	83.8%	<0.001 *
>65	67.5%		67.8%	
Sex				
Male	81.8%	0.255	78.9%	0.123
Female	85.8%		83.2%	
T stage				
T1–2	94.8%		94.0%	
T3	81.7%	<0.001 *	78.7%	<0.001 *
T4	63.3%	<0.001 *	56.8%	<0.001 *
N stage				
N1	91.3%		92.9%	
N2	85.1%	0.002 *	80.7%	<0.001 *
N3	69.1%	<0.001 *	61.4%	<0.001 *
Stage				
IB–II	93.1%		93.1%	
III	72.7%	<0.001 *	66.6%	<0.001 *
Lauren classification				
Non-intestinal	84.7%		81.4%	
Intestinal	80.8%	0.332	79.2%	0.704
Surgical extent				
STG	87.0%	<0.001 *	84.1%	0.001 *
TG	75.1%		72.8%	
Adjuvant treatment				
CCRT	84.4%	0.821	83.8%	0.097
Chemotherapy alone	82.3%		77.5%	
ANC (/µL)				
≤3448.5	82.7%	0.791	81.5%	0.567
>3448.5	84.1%		79.8%	
ALC (/µL)				
≤2074.0	84.1%	0.622	82.3%	0.344
>2074.0	82.6%		79.0%	
AMC (/µL)				
≤422.5	87.9%	0.001 *	86.7%	<0.001 *
>422.5	78.9%		74.6%	
PC (×10^3^/µL)				
≤240.5	85.3%	0.211	83.7%	0.058
>240.5	81.5%		77.5%	
NLR				
≤1.657	82.4%	0.921	80.5%	0.992
>1.657	84.3%		80.8%	
LMR				
≤4.956	80.0%	0.008^*^	76.6%	0.013 *
>4.956	86.7%		84.6%	
PLR				
≤117.6	84.7%	0.1449	82.8%	0.153
>117.6	82.1%		78.5%	

OS, overall survival; RFS, recurrence-free survival; STG, subtotal gastrectomy; TG, total gastrectomy; CCRT, concurrent chemoradiotherapy; ANC, absolute neutrophil count; ALC, absolute lymphocyte count; AMC, absolute monocyte count; PC, platelet count; NLR, neutrophil-to-lymphocyte ratio; LMR, lymphocyte-to-monocyte ration; PLR, platelet-to-lymphocyte ratio. Asterisks (*) indicate *p* < 0.005.

**Table 3 biomedicines-10-01565-t003:** Univariate Cox regression analysis for OS and RFS in all patients (*n* = 692).

		OS			RFS	
Variables	HR	95% CI	*p*-Value	HR	95% CI	*p*-Value
Age						
≤65	1.000			1.000		
>65	2.806	1.981–3.976	<0.001 *	2.089	1.462–2.983	<0.001
Sex						
Male	1.000			1.000		
Female	0.815	0.573–1.160	0.257	0.764	0.542–1.078	0.764
T stage			<0.001 *			<0.001 *
T1–2	1.000			1.000		
T3	2.844	1.730–4.675	<0.001 *	3.229	1.974–5.281	<0.001 *
T4	7.287	4.466–11.888	<0.001 *	7.658	4.712–12.447	<0.001 *
N stage			<0.001 *			<0.001 *
N1	1.000			1.000		
N2	2.105	1.293–3.426	0.003 *	2.933	1.769–4.861	<0.001 *
N3	4.599	2.895–7.304	<0.001 *	6.249	3.830–10.196	<0.001 *
Stage			<0.001 *			<0.001 *
IB–II	1.000			1.000		
III	4.518	3.010–6.782	<0.001 *	5.081	3.393–7.607	<0.001 *
Lauren classification			0.498			
Intestinal	1.000			1.000		
Non-intestinal	0.842	0.595–1.193	0.333	0.935	0.662–1.321	0.705
Surgical extent						
STG	1.000			1.000		
TG	2.204	1.572–3.090	<0.001 *	1.773	1.273–2.470	0.001 *
Adjuvant treatment						
CCRT	1.000			1.000		
Chemotherapy alone	1.040	0.742–1.457	0.821	1.319	0.949–1.835	0.100
ANC (/µL)						
≤3448.5	1.000			1.000		
>3448.5	1.047	0.747–1.467	0.792	1.100	0.792–1.527	0.568
ALC (/µL)						
≤2074.0	0.919	0.655–1.288	0.623	0.854	0.615–1.186	0.347
>2074.0	1.000			1.000		
AMC (/µL)						
≤422.5	1.000			1.000		
>422.5	1.802	1.270–2.555	0.001 *	2.023	1.434–2.853	<0.001 *
PC (× 10^3^/µL)						
≤240.5	1.000			1.000		
>240.5	1.241	0.884–1.741	0.213	1.374	0.987–1.913	0.060
NLR						
≤1.657	1.000			1.000		
>1.657	0.983	0.701–1.378	0.921	0.998	0.719–1.386	0.992
LMR						
≤4.956	1.583	1.122–2.234	0.009 *	1.514	1.085–2.113	0.015 *
>4.956	1.000			1.000		
PLR						
≤117.6	1.000			1.000		
>117.6	1.282	0.913–1.801	0.151	1.270	0.913–1.766	0.155

OS, overall survival; RFS, recurrence-free survival; HR, hazard ratio; CI, confidence interval; STG, subtotal gastrectomy; TG, total gastrectomy; CCRT, concurrent chemoradiotherapy; ANC, absolute neutrophil count; ALC, absolute lymphocyte count; AMC, absolute monocyte count; PC, platelet count; NLR, neutrophil-to-lymphocyte ratio; LMR, lymphocyte-to-monocyte ration; PLR, platelet-to-lymphocyte ratio. Asterisks (*) indicate *p* < 0.005.

**Table 4 biomedicines-10-01565-t004:** Multivariate Cox regression analysis for OS and RFS in all patients (*n* = 692).

		OS			RFS	
Variables	HR	95% CI	*p*-Value	HR	95% CI	*p*-Value
Age						
≤65	1.000			1.000		
>65	2.190	1.531–3.132	<0.001	1.444	1.001–2.084	0.049
T stage			<0.001			<0.001
T1–2	1.000			1.000		
T3	2.099	1.261–3.492	0.004	2.509	1.521–4.141	<0.001
T4	4.346	2.587–7.300	<0.001	4.717	2.829–7.865	<0.001
N stage			0.003			<0.001
N1	1.000			1.000		
N2	1.770	1.081–2.896	0.023	2.496	1.500–4.155	<0.001
N3	2.340	1.436–3.814	0.001	3.591	2.150–5.999	<0.001
Surgical extent						
STG	1.000					
TG	1.548	1.094–2.191	0.014			
AMC (/µL)						
≤422.5	1.000			1.000		
>422.5	1.674	1.180–2.376	0.004	1.908	1.650–2.695	<0.001

OS, overall survival; RFS, recurrence-free survival; HR, hazard ratio; CI, confidence interval; STG, subtotal gastrectomy; TG, total gastrectomy; AMC, absolute monocyte count.

**Table 5 biomedicines-10-01565-t005:** Multivariate Cox regression analysis for recurrence-free survival in patients with an absolute lymphocyte count > 2074.0/µL (*n* = 345).

Variables	Hazard Ratio	95% Confidence Interval	*p*-Value
T stage			<0.001
T1–2	1.000		
T3	2.285	1.202–4.344	0.012
T4	4.315	2.223–8.375	<0.001
N stage			0.001
N1	1.000		
N2	1.991	1.046–3.790	0.036
N3	3.374	1.751–6.499	<0.001
Adjuvant treatment			
CCRT	0.620	0.393–0.980	0.040
Chemotherapy alone	1.000		

CCRT, concurrent chemoradiotherapy.

**Table 6 biomedicines-10-01565-t006:** Meta-analyses that investigated the prognostic significance of systemic inflammatory markers in stomach cancer.

Author	Inclusion	Hazard Ratio (95% Confidence Interval)	Cut-Off
Kim et al. (2020) [20]	41 studies with 18,348 stage I–IV patients	High NLR, 1.605 (1.449–1.779) for OS	1.44–5.00
Du et al. (2021) [21]	36 studies with 8614 patients, inoperable	High NLR, 1.78 (1.59–1.99) for OS; 1.63 (1.39–1.91) for progression-free survival	0.4–5.0 (3.0 in 10 studies)
Ma et al. (2018) [19]	6 studies with 4908 stage I–IV patients	High LMR, 0.66 (0.54–0.82) for OS; not for DFS 0.71 (0.38–1.32)	3.15–5.15
Cao et al. (2020) [13]	28 studies with 15,617 stage I–IV patients	High PLR, 1.19 (1.08–1.33) for OS	108–305
Peng et al. (2022) [22]	17 studies with 3499 stage III–IV patients	High PLR, 1.429 (1.246–1.639) for OS; 1.47 (1.14–1.88) for DFS	107.7–284

NLR, neutrophil-to-lymphocyte ratio; OS, overall survival; LMR, lymphocyte-to-monocyte ratio; DFS, disease-free survival; PLR, platelet-to-lymphocyte ratio.

## Data Availability

Not applicable.

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
