# Peer review of "Clinical Significance of Preoperative Hematological Parameters in Patients with D2-Resected, Node-Positive Stomach Cancer"

_biomedicines, 2022, doi:10.3390/biomedicines10071565_

Round 1

Reviewer 1 Report

The authors reported “Clinical significance of preoperative hematologic parameters in

patients with D2-resected, node-positive stomach cancer”. The authors concluded that findings of this study could be useful for radiation oncologists investigating the role of adjuvant radiotherapy after D2 resection.

There are some problems in this paper, I think.

  1. The conclusion of this study is based on Cox model in subgroup-patient (Table5). I think this is a bit forcible.
  2. In this study, continuous variables were converted into categorical variables based on the mean values. Is it appropriate?
  3. There are not purpose and conclusion of this study in abstract part. It is not reader-friendly, I think.
  4. It is unclear why the authors conduct this study. In Introduction part, the background of this is unclear.
  5. The box plot presentation seems better in Figure 1, compared to error-bar presentation.
  6. There were significant differences in patient characteristics between the different adjuvant treatment groups owing to the retrospective study design. I recommend propensity matching.
  7. The authors presented multivariate analyses in subgroup in Supplemental Table1. However, these evaluations seem ambiguous. What is the clinical significance?
  8. There are many supplemental Figures. These evaluations also seem ambiguous. It is unclear why these evaluations add depth of this study. Furthermore, the authors should put these supplemental Figures together on word-spread sheet to make them reviewer-friendly.
  9. The authors should explain more about what this result is useful in routine clinical practice.

Reviewer 2 Report

Well written manuscript with interesting findings, though adjuvant treatment with chemo-radiotherapy is largely reserved for margin positive patients where further surgery is contraindicated, or for patients who have undergone suboptimal lymph node dissection.

It would be good to present more data on the surgery outcomes: eg. margin positivity (R0/1/2 status), number of LNs harvested during surgery, complication rates, duration from surgery to start of adjuvant treatment, adjuvant treatment related toxicities, to make better sense of the data. Were there any patients who had neo-adjuvant treatment prior to surgery?

Also, as the authors postulate that the immuno response may be one of the explainations for their conclusions, it would be good to know what the authors opinion of immuno therapy is and how this could be included in future studies looking at hematologic parameters for patients undergoing surgery.
